# Bioindicator snake shows genomic signatures of natural and anthropogenic barriers to gene flow

**Damian C. Lettoof**[1]*, **Vicki A. Thomson**[2], **Jari Cornelis**[1], **Philip W. Bateman**[1], **Fabien Aubret**[3,4], **Marthe M. Gagnon**[4], **Brenton von Takach**[4,5]

1 Behavioural Ecology Lab, School of Molecular and Life Sciences, Curtin University, Bentley, Western Australia, Australia, 2 School of Biological Sciences, University of Adelaide, Adelaide, South Australia, Australia, 3 Station d'Ecologie Théorique et Expérimentale, CNRS, Moulis, France, 4 School of Molecular and Life Sciences, Curtin University, Bentley, Western Australia, Australia, 5 Research Institute for the Environment and Livelihoods, Charles Darwin University, Darwin, Northern Territory, Australia

☯ These authors contributed equally to this work.
* damian.lettoof@postgrad.curtin.edu.au

**Data Availability Statement:** The data is stored in the public database BioStudies, reference:S-BSST646. https://www.ebi.ac.uk/biostudies/studies/S-BSST646. Data includes:Morphological data and individual heterozygosities for all snakes

## Abstract

Urbanisation alters landscapes, introduces wildlife to novel stressors, and fragments habitats into remnant 'islands'. Within these islands, isolated wildlife populations can experience genetic drift and subsequently suffer from inbreeding depression and reduced adaptive potential. The Western tiger snake (*Notechis scutatus occidentalis*) is a predator of wetlands in the Swan Coastal Plain, a unique bioregion that has suffered substantial degradation through the development of the city of Perth, Western Australia. Within the urban matrix, tiger snakes now only persist in a handful of wetlands where they are known to bioaccumulate a suite of contaminants, and have recently been suggested as a relevant bioindicator of ecosystem health. Here, we used genome-wide single nucleotide polymorphism (SNP) data to explore the contemporary population genomics of seven tiger snake populations across the urban matrix. Specifically, we used population genomic structure and diversity, effective population sizes ($N_e$), and heterozygosity-fitness correlations to assess fitness of each population with respect to urbanisation. We found that population genomic structure was strongest across the northern and southern sides of a major river system, with the northern cluster of populations exhibiting lower heterozygosities than the southern cluster, likely due to a lack of historical gene flow. We also observed an increasing signal of inbreeding and genetic drift with increasing geographic isolation due to urbanisation. Effective population sizes ($N_e$) at most sites were small (< 100), with $N_e$ appearing to reflect the area of available habitat rather than the degree of adjacent urbanisation. This suggests that ecosystem management and restoration may be the best method to buffer the further loss of genetic diversity in urban wetlands. If tiger snake populations continue to decline in urban areas, our results provide a baseline measure of genomic diversity, as well as highlighting which 'islands' of habitat are most in need of management and protection.

without food, Raw SNP data for all snakes used in analysis, as provided by DARTseq, and site and coordinates of each snake used in analysis. Data will be available upon publication.

**Funding:** DCL received funding for this research from the Holsworth Wildlife Research Endowment, https://www.ecolsoc.org.au/awards/holsworth/. VAT received funding for this research from the Australian Research Council, LP160100189 and DE180100624, https://www.arc.gov.au/. The funders had no role in study design, data collection and analysis, decision to publish, or preparation of the manuscript.

**Competing interests:** No authors have competing interests.

# Introduction

Urbanisation, the anthropogenic transformation of natural ecosystems via the growth of cities [1], introduces wildlife to a myriad of stressors such as dynamic availability of resources [2], pollution [3, 4], novel environments [5, 6] and human disturbance [7]. These novel stressors affect the behaviour, physiology and health of wildlife [8], and consequently create strong selection pressures driving evolution [9–11]. Additionally, urban development fragments habitats resulting in remnant patches becoming isolated islands in a matrix of urbanisation [12–14]. The less-mobile, philopatric or more habitat-specialist species may persist only in these islands and not the surrounding matrix, and thus face the random genetic pressures inherent to isolated populations of such species with reduced or non-existent gene flow between subpopulations [15, 16].

Isolated populations are expected to experience increased levels of genetic drift–stochastic loss of alleles through time–and differentiation, in conjunction with reduced genetic diversity within populations [16]. In instances where the remnant population is small, a reduction in genetic diversity can potentially lead to signs of inbreeding depression [17, 18]. This inbreeding depression, the overexpression of deleterious recessive alleles in homozygotes, can also lead to a reduction in individual fitness [19, 20], while genetic drift can lead to reduced adaptive potential [16, 21]. Consequently, urban 'island' populations are in a fitness and adaptation arms race against the constant stressors of urbanisation.

The Australian city of Perth is built on the Swan Coastal Plain (SCP); a bioregion characterised by banksia woodland on sand dunes, intersected by north-south connected chains of ephemeral wetlands. Since 1850, urban development and agriculture in the SCP led to the loss 70% of the original wetland area [22], with most of the remaining wetlands suffering from severe degradation. Tiger snakes (*Notechis scutatus*) are a ~ 1 m elapid snake restricted to the cooler, wetter climates of Australia [23]; they prefer wetland habitats on the mainland, yet numerous populations exist on very dry off-shore islands [24]. Tiger snakes were once considered under threat of extinction for a number of reasons, but primarily the destruction and degradation of wetland habitats due to urbanisation [25]. Although tiger snakes are still regionally common, there is anecdotal evidence of their decline in some cities and on some islands. For example, Eastern tiger snakes (*N. scutatus scutatus*) were once locally abundant in greater Sydney but now only persist on the outskirts of the city [26].

The loss of tiger snakes across some of their distribution is not enough to label the species with conservation concern; however, snakes can be useful indicators of ecosystem health [27–29]. Perturbation of their populations can thus inform land managers of the integrity of the environment. Prior to urbanisation, Western tiger snakes (*N. scutatus occidentalis*) in the SCP likely moved among ephemeral wetlands as these environments dried throughout the warmer months, but now populations only persist around, and appear restricted to, several large lakes and river edges with sufficient fringing vegetation (DL pers. obs.). Despite persisting in these fragmented wetland habitats thus far, they are exposed to and bioaccumulate a suite of contaminants that likely contribute to poorer health and decreased survival of individuals [30–32]. However, the degree to which small population sizes, geographic isolation and inbreeding effects contribute to population health in these urban and peri-urban populations has not yet been investigated.

To address these knowledge gaps, we assessed the population structure and patterns of genomic diversity in tiger snake populations persisting in and around the city of Perth in Western Australia. We included a 'recently-introduced' off-shore island population to allow for a comparison with the genomic structure of a true island population. Analyses included calculating and comparing the effective population sizes across populations to explore the

impacts of genetic drift and isolation, and compare individual heterozygosity to a body condition index [33, 34] to investigate the potential relationship between fitness and heterozygosity. We predicted that the major river systems that divide Perth (Fig 1) would be barriers to gene flow between the northern and southern localities, as observed in other species [35], and that those populations more isolated by urbanisation would show lower levels of genomic diversity, higher pairwise genomic differentiation and stronger signals of inbreeding, which would correlate with lower fitness. This study explores the contemporary population genomics of a large elapid persisting in wetlands threatened by ever-increasing urbanisation, and highlights which populations are at risk of extirpation in the future.

## Materials and methods

### Study sites and sample collection

We sampled 150 tiger snakes from six wetlands around Perth: Loch McNess (*n* = 22; within Yanchep National Park, 31˚32'45" S, 115˚40'50" E), Lake Joondalup (*n* = 23; 31˚45'37" S, 115˚47'36" E) and Herdsman Lake (*n* = 57; 31˚55'14", S 115˚48'18" E), located north of the Swan/Canning River system, and Bibra Lake (*n* = 29; 32˚05'33", S 115˚49'31" E), Kogolup Lake (*n* = 10; 32˚07'40", S 115˚50'05" E) and Black Swan Lake (*n* = 9; 32˚28'32", S 115˚46'22" E) located south of the Swan/Canning River system. These study sites represent the northern extremity of the Western tiger snake distribution (Fig 1). We also collected nine samples from Carnac Island, approximately 7 km off-shore (Fig 1). Carnac Island is a small freshwater-devoid island (19 ha) with the tiger snake population thought to originate from human introduction approximately 90 years ago, with the suspected source population coming from the nearby mainland [24, 36]. Kogolup Lake, Black Swan Lake and Carnac Island were surveyed less than the other sites (a few days compared with several weeks), which resulted in lower sample sizes for these sites.

We took ventral scale clips from each snake collected from the six mainland sites during September–October 2020. We stored individual scales in 95% ethanol at -20˚C until extraction. The Carnac Island tiger snakes also had scale clips collected from April 2018–January 2020 that were stored in 95% ethanol at 4˚C until extraction. We also included two additional snakes from eastern Australia, > 2000 km from the Perth populations, as outgroup samples. Tiger snakes from the mainland sites were sampled under Curtin University's Animal Research Ethics approval: ARE2018-23 and ARE2020-6; and Western Australia's Department of Biodiversity, Conservation and Attractions permits: FO25000149 and FO25000294-2. The nine tiger snakes from Carnac Island were sampled under the University of Adelaide's Animal Research Ethics permits: S-2016-111; and Western Australia's Department of Biodiversity, Conservation and Attractions permits: 01-000069-3, FO25000008, and FO25000008-2.

### DNA extraction, sequencing and bioinformatics pipeline

Tissue samples were sent to the DArTseq laboratory in Canberra, ACT for DNA extraction, library preparation and double-digest restriction-site associated DNA next-generation sequencing. Briefly, the library preparation consisted of DNA digestion using the restriction enzymes *PtsI* and *HpaII*, as these enzymes had previously been used for tiger snake RAD-seq library preparation. Following digestion, adapter ligation and PCR amplification, DNA libraries were sequenced on a single lane of an Illumina Hiseq 2500 platform. The DArTseq proprietary bioinformatic pipeline [38] was used to demultiplex, clean, and filter reads, then map reads to the *Notechis scutatus* reference genome (NCBI PRJ: PRJEB27871) and call single-nucleotide polymorphisms (SNPs). Detailed methods covering DArTseq library preparation, sequencing, read filtering and SNP calling have been provided in previous publications

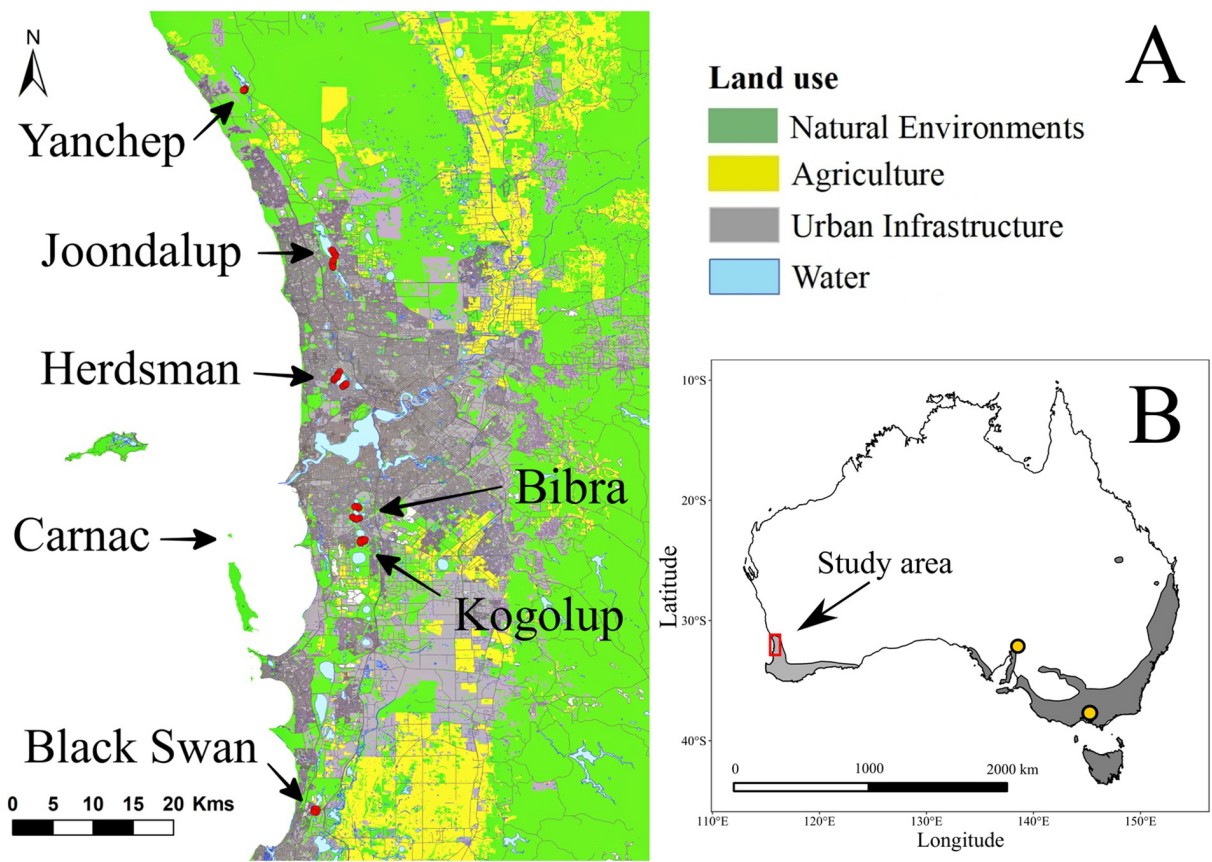

**Fig 1. Map the studied populations of *Notechis scutatus occidentalis* and land-use of Perth, Western Australia.** Red points represent individual Western tiger snakes, and yellow points represent Eastern tiger snakes (*Notechis scutatus scutatus*). Grey shading represents the current distribution extent of the species (light = Western, dark = Eastern; modified from the IUCN Red List of Threatened Species [37]). Land-use was classified by the 2016 Australian Land Use and Management Classification (Australian Bureau of Agricultural and Resource Economics and Sciences, Canberra. CC BY 3.0.).

[39, 40]. We received a SNP-by-sample matrix consisting of 161 samples and 22542 SNPs, which was read into R v4.0.3 for subsequent SNP filtering and analysis.

## SNP filtering

We used a custom R script to prune unwanted SNPS and retain SNPs of interest. To remove potential bias due to sequencing or genotyping errors, we retained SNPs with total read depth > 10 and < 100, as well as those with high DArTseq reproducibility scores (no SNP < 100% reproducible in technical replicates). Reads that did not map to the reference genome in the DArTseq pipeline were also discarded. To account for bias due to linkage dis-equilibrium [41], we filtered out SNPs in close proximity to one another, by retaining just one SNP from each RAD locus. We also retained only SNPs that were genotyped in a high propor-tion of samples (callrate > 0.95), had a minor allele count ≥ 3, and observed heterozygosity < 0.6. Finally, we removed any sample that had > 20% missing data. This pro-duced a large, high-quality SNP-by-sample matrix, with a low overall level of missing data (1%). Once filters had been applied we retained 4688 SNPs from 159 Western and two Eastern tiger snake individuals.

As a preliminary measure to investigate whether our filtered dataset was appropriate for inferring relationships between individuals and populations, we created and visually inspected a hierarchical clustering dendrogram based on Nei' genetic distance (S1 Fig). All individuals fell within their sampled localities, and populations clustered in line with expected geographic distances and landscape barriers, including the two individuals from eastern Australia, suggesting that the retained SNP dataset was adequate for further analysis. The eastern individuals were then removed from subsequent analyses. As further confirmation of data integrity, we investigated pairwise kinship coefficients within each population. To ensure relatedness between individuals was not high enough to impact our data analysis or interpretation, we calculated pairwise kinships via the 'beta.dosage' function of the *hierfstat* package (version 0.7) [42, 43]. Four pairwise values from a total of 2604 comparisons had kinships > 0.25 but < 0.3 (commonly values of 0.25 indicate full siblings), therefore no individuals were removed.

### Regional population structure

To investigate genomic distance between individuals we calculated the individual pairwise genetic distances using the 'prevosti.dist' function in the *poppr* package (version 2.9.1) [44]. We visualised these distances via the 'cmdscale' function, plotting the first two dimensions of the genetic distance matrix in a multidimensional scaling plot–where the distances between points are approximately equal to the dissimilarities. We then calculated pairwise population genomic differentiation values ($G''_{ST}$) among all sampling localities using the 'pairwise_Gst_Hedrick' function of the *mmod* package (version 1.3.3) [45], and obtained *p* values for all population pairs using the *StAMPP* package (version 1.6.2) [46]. The $G''_{ST}$ metric is an $F_{ST}$ analogue representing a standardised measure of allelic isolation that corrects for the number of subpopulations being considered.

To explore genomic structure and assess potential gene flow among populations, we searched for genomic groups using the TESS3 algorithm–a spatially explicit ancestry estimation model from the *tess3r* package (version 1.1) [47, 48]. The 'tess3' function incorporates the latitudes and longitudes of each sampled individual to account for the influence of isolation-by-distance on ancestry coefficients, with large drops or plateaus in the scree plot identifying useful values of *K* for inference of population genetic structure (S2 Fig). We considered *K* values of 2, 3 and 4 most useful for describing the high-level genetic structure across the region, as these values showed the largest reductions in cross-entropy, with a plateau starting to form at *K* > 4. For finer-scale investigation of population structuring we also plotted ancestry coefficients of *K* = 5, 6 and 7 (S3 Fig). To confirm whether there SNPs under putative selection were driving patterns of population structure, we conducted a genome scan for outlier loci using the 'pvalue' function. This outlier test uses overall differentiation to discern if a portion of SNPs have greater allele frequency differences than expect from a neutral distribution [49]. We used *K* = 4 as we considered this the most useful for inference, and a Benjamini-Hochberg correction to achieve a false discovery rate of one in 10 000.

To investigate isolation-by-distance, we calculated the multilocus spatial autocorrelation for the mainland, and subset of northern and southern populations of snakes, respectively. The spatial autocorrelation analysis was conducted in GENALEX (version 6.5) add-on in Excel [50, 51].

### Population genomic diversity

To investigate patterns of within population genomic diversity, we calculated standard genetic diversity metrics for all *a-priori* populations using the GENALEX in Excel. The diversity metrics included mean values for the number of alleles ($N_A$), effective number of alleles ($A_E$),

information index ($I$), observed heterozygosity ($H_O$), expected heterozygosity ($H_E$), and the fixation index (Wright's inbreeding coefficient, $F_{IS}$). We also calculated the number of private alleles in each population as an additional measure of genetic distinctiveness. To standardise the number of private alleles for different sample sizes among populations we bootstrapped private allele calculations by resampling nine individuals per population 100 times, and taking the mean and SE of all bootstraps.

## Heterozygosity-fitness correlations

As individual heterozygosity is often related to individual fitness [21, 52, 53], we modelled the relationship between individual heterozygosity and body condition of all genotyped individuals. Previous work has shown that approximately 50% of the variation in snake body condition estimates results from stored body fat, while organ mass such as muscle and liver account for the remainder [54, 55]. To determine body condition we calculated a scaled mass index (SMI) for each snake using the formula: $SMI = Wi\left(\frac{L0}{Li}\right)b_{SMA}$ where $W_i$ and $L_i$ are the weight and snout-vent length (SVL) of individuals, $L_0$ is the arithmetic mean length of all sampled individuals, and $b_{SMA}$ is the scaling exponent estimated by the standardised major axis regression of mass on length of all sampled individuals [56, 57]. We consider the SMI to be an estimate of fitness with higher values corresponding to fitter individuals [33]. To increase accuracy of body condition calculations, we excluded snakes with obvious gastric food items or pregnancy. We also excluded Carnac Island snakes as this population was sampled in summer, a time when these individuals potentially have low body condition (from low prey and water availability). Based on 91 snakes, $L_0$ used in the equation was 757.1 mm, and the $b_{SMA}$ was 2.98. To explore evidence of heterozygosity-fitness correlations we ran a general linear mixed model (GLMM) using the 'glmer' function of the *lme4* package (version 1.25) [58]; with SMI as the response variable, individual heterozygosity and site as fixed predictor variables and to account for sex-biased differences, sex as a random factor. We used a histogram of the model residuals to confirm the assumptions of linearity.

In addition, we compared body condition with $g^2$ –a proxy for identity disequilibrium (inbreeding) [53, 59]. A $g^2 = 0$ means no variance of inbreeding in the sample. With the *inbreedR* package (version 0.3.2) [60], we used the 'r2_wf' function to calculate the expected correlation between inbreeding level (*f*) and body condition, and the 'r2_hf' to calculate the correlation between inbreeding level (*f*) and individual heterozygosity.

## Effective population size

The effective population size ($N_e$) of a site represents the estimated number of breeding adults in a single generation of an ideal population that shows the same degree of genetic diversity as the measured population [61, 62]. Theoretically, small estimates of $N_e$ reflect small, isolated populations suffering increased drift and lower fitness (e.g. through inbreeding), whereas large values of $N_e$ reflect large and genetically diverse populations [63, 64]. An effective population size can also be used to estimate adult census size [65]. To estimate $N_e$ of each population we used the widely-used linkage disequilibrium method, as it is considered one of the most suitable for single-sample datasets [62]. $N_e$ estimates were calculated using the NEESTIMATOR v2.1 [66].

## Results

### Regional population structure

We found expected levels of genomic differentiation between populations, based on geographic distance and landscape barriers between sites (Table 1). Carnac Island was the most

**Table 1. Pairwise values of population genomic differentiation ($G''_{ST}$) of *Notechis scutatus occidentalis* around Perth, Western Australia.**

| Site | Yanchep | Lake Joondalup | Herdsman Lake | Bibra Lake | Kogolup Lake | Black Swan Lake | Carnac Island |
|---|---|---|---|---|---|---|---|
| **Yanchep** ($n = 22$) | 0 | | | | | | |
| **Lake Joondalup** ($n = 23$) | 0.14 | 0 | | | | | |
| **Herdsman Lake** ($n = 57$) | 0.24 | 0.18 | 0 | | | | |
| **Bibra Lake** ($n = 29$) | 0.24 | 0.20 | 0.21 | 0 | | | |
| **Kogolup Lake** ($n = 10$) | 0.22 | 0.19 | 0.20 | 0.07 | 0 | | |
| **Black Swan Lake** ($n = 9$) | 0.26 | 0.22 | 0.23 | 0.14 | 0.13 | 0 | |
| **Carnac Island** ($n = 9$) | 0.40 | 0.38 | 0.38 | 0.31 | 0.29 | 0.29 | 0 |

All non-zero values are highly significant ($p < 0.001$).

differentiated from other populations (pairwise $G''_{ST}$ = 0.29–0.40). The most geographically distant pair of sites, Yanchep and Black Swan, exhibited a moderate level of differentiation ($G''_{ST}$ = 0.26). Black Swan Lake was less differentiated from Kogolup and Bibra Lakes ($G''_{ST}$ = 0.13–14) than Yanchep was to Herdsman Lake ($G''_{ST}$ = 0.23), despite these pairs of locations being a similar geographic distance from one another (~45 km). Analysis of spatial autocorrelation identified significant values of the autocorrelation coefficient $r$ persisting for distances up to 38 km (S4 Fig). We found a difference in the decay of spatial autocorrelation between the northern and southern clusters, with the northern cluster intercepting $r = 0$ at about 30 km and the southern cluster intercepting $r = 0$ at about 12 km.

The outlier test identified 131 (2.8% of 4688) SNPs as being under putative selection. These loci demonstrate significantly higher or lower among-population genetic differentiation than expected under neutrality, many of which are possibly driven by the differentiation between Carnac Island and the mainland populations. While this small number of SNPs is unlikely to have substantially influenced our observed population structure, these loci could be responsible for influencing fitness under differing environmental conditions, and thus provide a basis for further study.

The multidimensional scaling plot (Fig 2) showed four main population clusters of tiger snakes in the Perth region, representing (1) Carnac Island, (2) Herdsman Lake, (3) Yanchep and Joondalup lakes, and (4) all three lakes (Bibra, Kogolup and Black Swan Lakes) on the southern side of the Swan River. 24.5% of the total variation was explained by the first two axis. The Carnac Island cluster separated strongly from the other three clusters on both the first and second coordinate.

Investigation of the cross-entropy scree plot produced using the *tess3r* package indicated a likely number of ancestral lineages at $2 \leq K \leq 4$ (S2 Fig). Plotting ancestry coefficients for all individuals at each of these $K$ values highlights the genomic separation of the northern lakes from the southern lakes ($K = 2$), with Herdsman Lake separating from the two other northern lakes at $K = 3$, and Carnac Island separating from the southern lakes at $K = 4$ (Fig 3). With finer-scale population splitting, Lake Joondalup separates from the northern lakes ($K = 5$) while Kogolup Lake and Black Swan Lake share ancestry with Carnac Island (S4 Fig). At $K = 6$, Black Swan Lake separates from the southern lakes, and Kogolup Lake is only recognised as a unique different population at $K = 7$ (at which point all *a-priori* populations cluster separately). S5 Fig visualises the hierarchical population genomic structure for $K$ values between 2 and 6.

## Genomic diversity and health

Genomic diversity was generally lower in populations north of the river than in populations south of the river, with the Carnac Island population having the highest heterozygosity of all

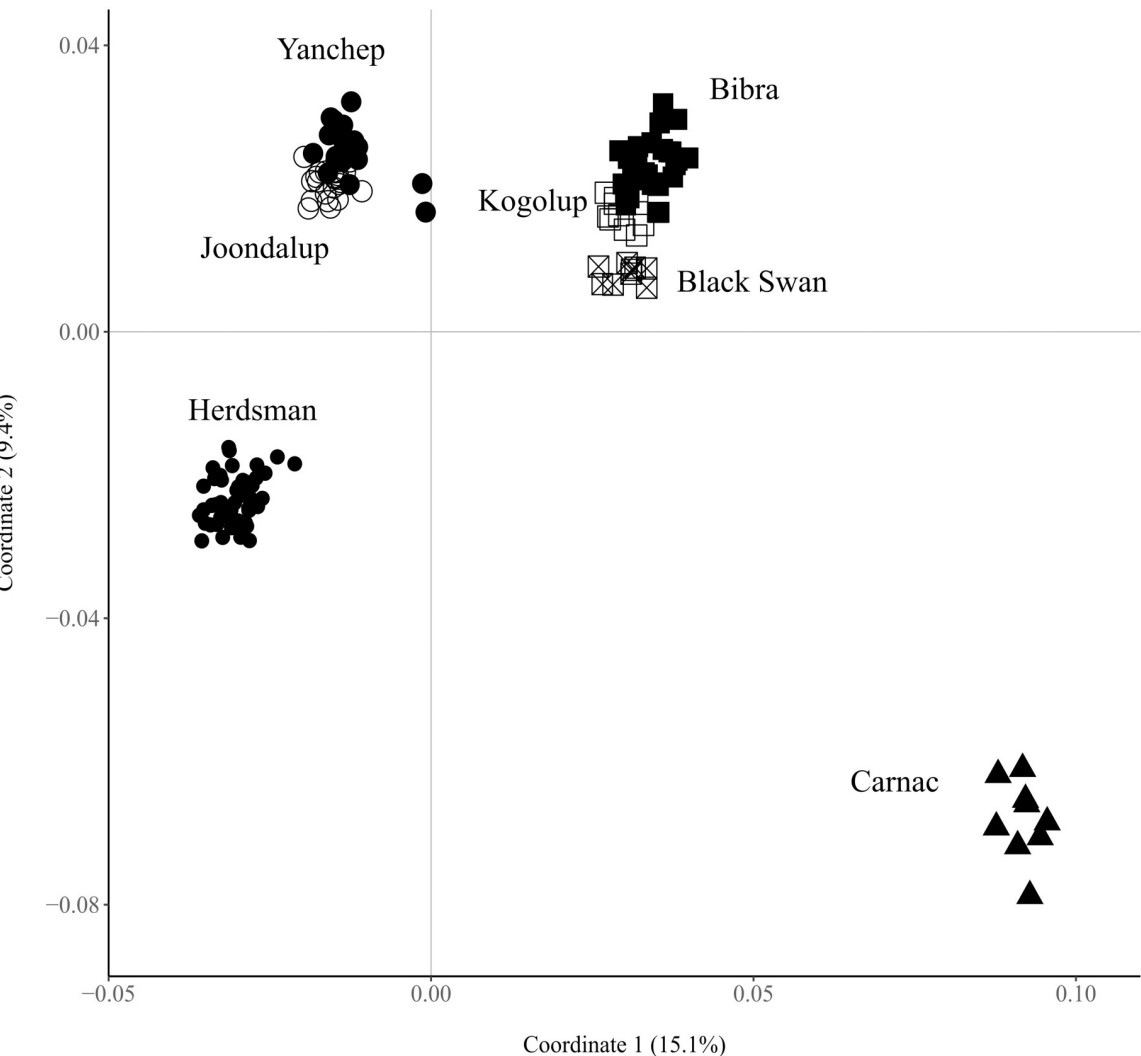

**Fig 2. Multidimensional scaling plot of genetic distance between *Notechis scutatus occidentalis* individuals from Perth, Western Australia.** Each population has been given a unique icon. Axis one explained 15.1% and axis one explained 9.4% of the total variance.

studied populations (Table 2). Observed heterozygosity, specifically, was 25–33% lower in the northern populations. Observed heterozygosity did not differ from expected heterozygosity in most populations, although Carnac Island showed a slightly lower $H$o (0.13) than $H$e (0.14). Carnac Island had the highest relative number of private alleles (328). The signal of inbreeding (i.e. $F_{IS}$ values) increased with the level of geographic isolation in *a-priori* populations; the true island population (Carnac Island) had the highest $F_{IS}$ value (0.05), with the mainland populations most impacted by urbanisation having higher $F_{IS}$ values (0.04–0.02) than less disturbed populations ($\leq 0$). While these values are close to zero, they likely reflect genome-wide patterns and differences between populations, with low values not unexpected when using a low minor allele frequency threshold.

The three northern populations, Yanchep, Lake Joondalup and Herdsman Lake, had lower mean body conditions compared to the southern populations (Fig 4). The GLMM ($r^2 = 0.43$) showed no evidence for a significant relationship between individual heterozygosity and body condition (F: 0.51, df; 1, $p = 0.48$); however, there was a strong effect of site (F: 3.22, df: 5,

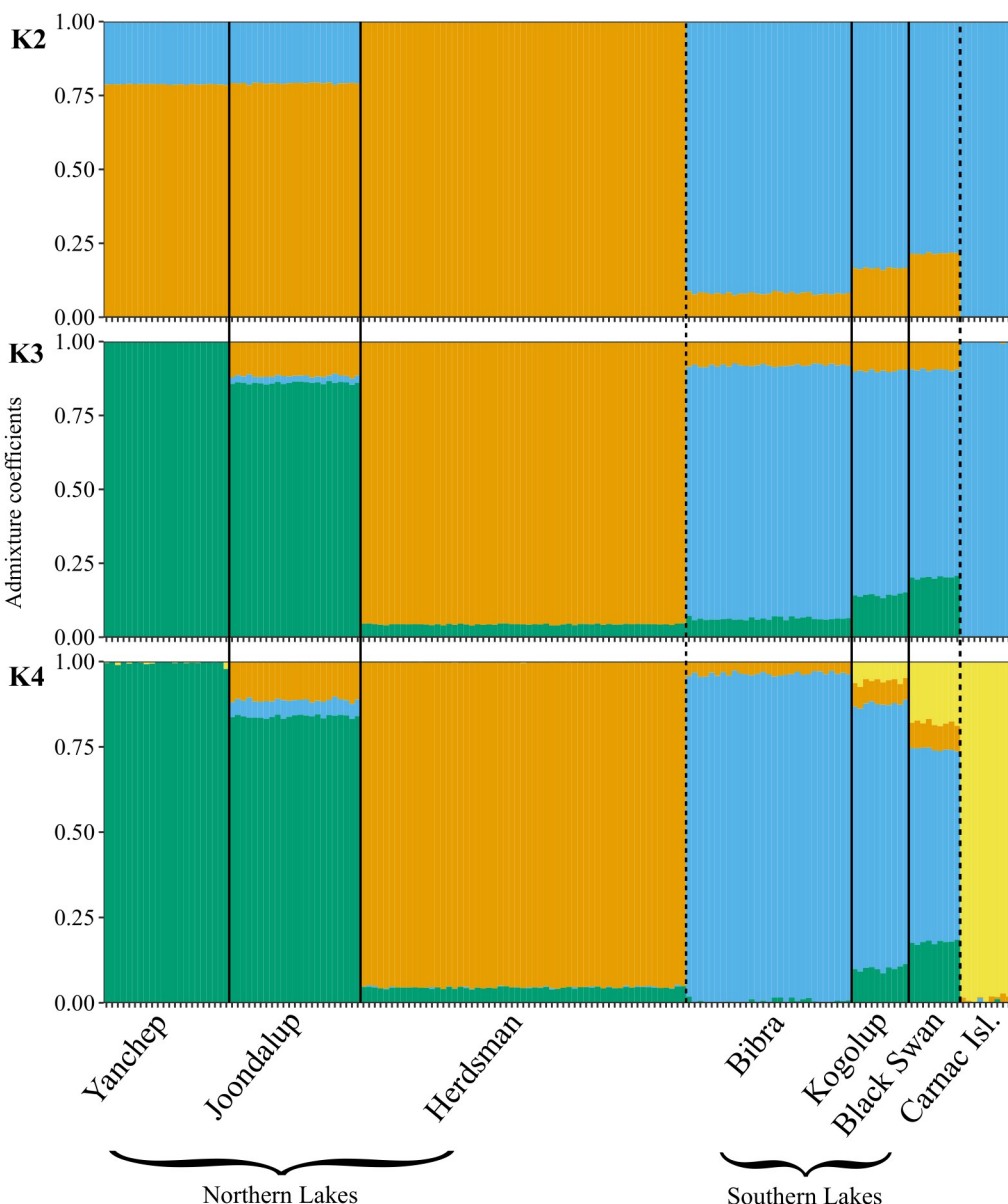

**Fig 3. Admixture bar plot comparing population structure in *Notechis scutatus occidentalis* from Perth, Western Australia.** Each tick mark on the x-axis represents an individual snake, which are grouped by sampling locations. The dashed line represents the biogeographic barrier of the Swan/Canning rivers separating the northern and southern sampling localities. The y-axis represents the fraction of individuals' genome that originates from a particular ancestral population, each of which has been given a unique colour.

$p = 0.01$). A Tukey HSD test found the greatest, and only significant, difference in body condition was between Herdsman Lake and Black Swan Lake snakes ($p = 0.03$; S1 Table). Of the snakes with body condition data, the inbreeding among loci was significantly different from zero ($g^2 = 0.025 \pm 0.003$ S. E., $p = 0.01$). We found a high correlation between inbreeding level (*f*) and heterozygosity ($r^2 = 0.94$), strongly suggesting that heterozygosity is a good proxy for inbreeding. Furthermore, we found no correlation between inbreeding level (*f*) and body condition ($r^2 = 0.07$).

**Table 2. Genetic diversity estimates of seven populations of *Notechis scutatus occidentalis* around Perth, Western Australia.**

| Site | $N_A$ | $A_E$ | $I$ | $Ho$ | $He$ | $F_{IS}$ | Private alleles |
|---|---|---|---|---|---|---|---|
| **Yanchep (*n* = 22)** | 1.32 (0.01) | 1.14 (<) | 0.13 (<) | 0.08 (<) | 0.08 (<) | 0.00 (<) | 55.79 (0.89) |
| **Lake Joondalup (*n* = 23)** | 1.33 (0.01) | 1.15 (<) | 0.14 (<) | 0.09 (<) | 0.09 (<) | 0.02 (<) | 56.27 (0.53) |
| **Herdsman Lake (*n* = 57)** | 1.39 (0.01) | 1.15 (<) | 0.14 (<) | 0.09 (<) | 0.09 (<) | 0.04 (<) | 69.49 (0.73) |
| **Bibra Lake (*n* = 29)** | 1.45 (0.01) | 1.20 (<) | 0.19 (<) | 0.12 (<) | 0.12 (<) | 0.03 (<) | 81.73 (0.86) |
| **Kogolup Lake (*n* = 10)** | 1.39 (0.01) | 1.19 (<) | 0.18 (<) | 0.12 (<) | 0.12 (<) | -0.02 (<) | 68.17 (0.59) |
| **Black Swan Lake (*n* = 9)** | 1.38 (0.01) | 1.19 (<) | 0.18 (<) | 0.12 (<) | 0.12 (<) | -0.02 (<) | 111.30 (0.46) |
| **Carnac Island (*n* = 9)** | 1.41 (0.01) | 1.25 (0.01) | 0.21 (<) | 0.13 (<) | 0.14 (<) | 0.05 (0.01) | 328.03 (0.97) |

Presented as mean values across all SNPs; (<), standard error < 0.01; $Na$, no. of alleles; $A_E$, effective number of alleles; $I$, information index; $Ho$, observed heterozygosity; $He$, expected heterozygosity; $F_{IS}$, fixation index (Wright's inbreeding coefficient).

## Effective population size

We found the largest $N_e$ at Herdsman Lake (95% CI $N_e$ = 152–162), with the smallest effective population size at Black Swan Lake (95% CI $N_e$ = 24–26; Table 3). The effective population size at Kogolup Lake was 'infinite', likely due to small sample size preventing calculation of $N_e$. The

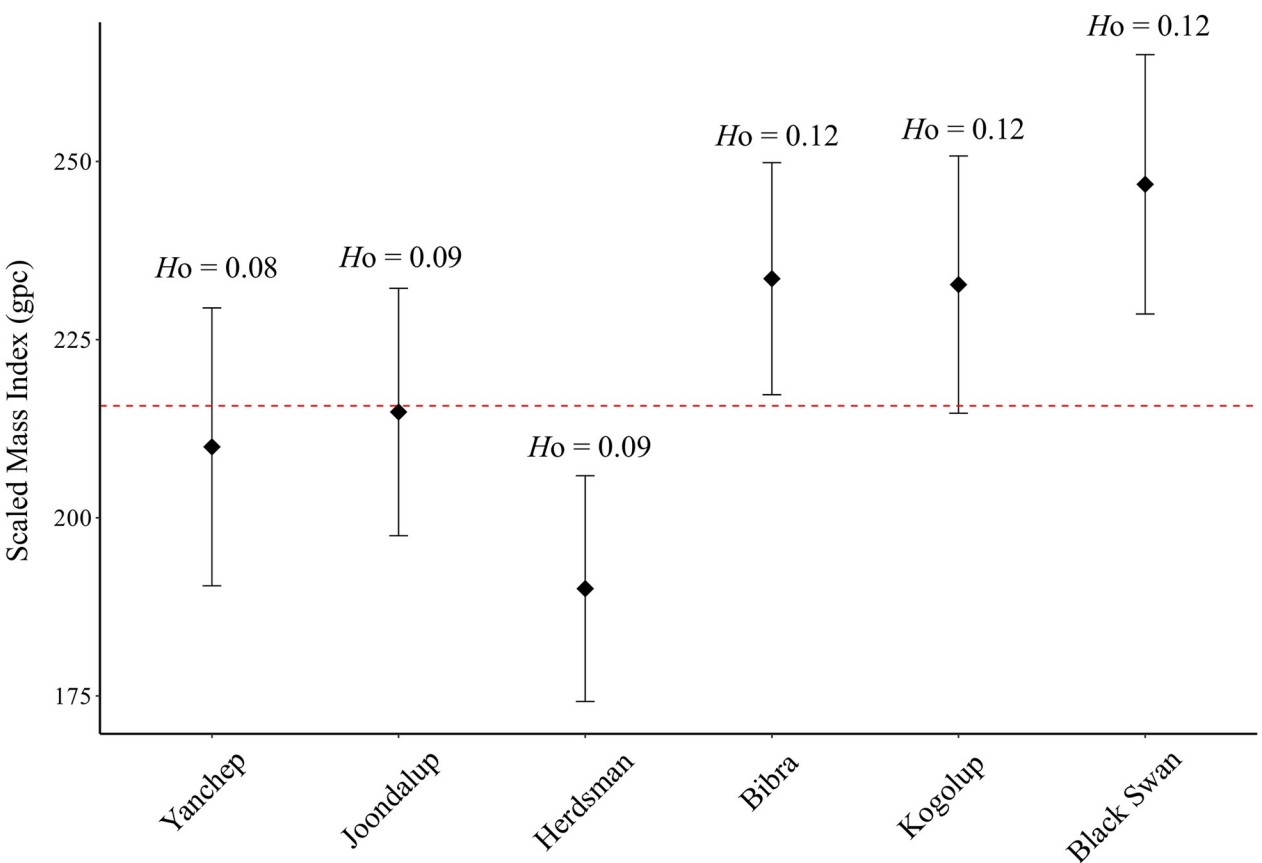

**Fig 4. Mean body condition of mainland *Notechis scutatus occidentalis* populations around Perth, Western Australia.** Body condition is presented as scaled mass index (SMI). gpc, grams per cm; filled diamonds are the population mean; error bars represent 95% confidence intervals; dashed line is mean SMI of 215.7 gpc at the mean population SVL of 757.1 mm; $Ho$ is the mean observed heterozygosity for each population.

**Table 3. Effective population size ($N_e$) of seven populations of *Notechis scutatus occidentalis* around Perth, Western Australia.**

| Site | $N_e$ | Overall $r^2$ | Parametric 95% CI |
|---|---|---|---|
| **Yanchep** | 92 | 0.056 | 86–99 |
| **Lake Joondalup** | 110 | 0.053 | 104–118 |
| **Herdsman Lake** | 156 | 0.021 | 152–162 |
| **Bibra Lake** | 84 | 0.042 | 82–87 |
| **Kogolup Lake** | INF | 0.138 | 3581–INF |
| **Black Swan Lake** | 25 | 0.170 | 24–26 |
| **Carnac Island** | 90 | 0.168 | 82–101 |

95% CI represent both upper and lower confidence intervals; INF, 'infinite', usually a result of not enough information to obtain a reliable estimate. Overall $r^2$ represents a composite measure of average linkage-disequilibrium across all pairs of loci.

$r^2$ –a sample-size bias corrected value of linkage-disequilibrium across all loci pairs–was lower in the northern populations than in the southern populations.

## Discussion

### Regional population structure

Population genomic analysis generally supported our *a-priori* predictions. At the highest hierarchical level, we identified population clusters that aligned with geographic regions north and south of the Swan/Canning River systems (hereafter the northern and southern cluster). Broadly, our findings reflect the results of Ottewell, Pitt [35], who found that the quenda (*Isoodon fusciventer*), a small marsupial persisting in Perth bushland patches, showed a similar pattern of genomic differentiation on either side of the major rivers. Although tiger snakes are capable swimmers [67, 68] they are unlikely to swim across the width of the Swan/Canning rivers, and these rivers have likely been historic natural landscape barriers reducing gene flow among tiger snake populations in this region.

Finer-scale patterns identified populations at Herdsman Lake and Carnac Island as being more genetically distinct within these broader clusters, likely due to a combination of genetic drift and isolation. At $K = 3$, Herdsman Lake is recovered as a distinct cluster, potentially due to isolation from a sea of urban infrastructure; at $K = 4$, Carnac Island is distinct, likely due to its isolation by 7 km of ocean. We also observed increased genomic distinction in populations reflecting the history and level of surrounding urbanisation. Remnant patches of habitat provide connectivity and allow for persistence of wildlife in urban areas [35, 69]; however, as Western tiger snakes prefer wetland habitats, they are unlikely to disperse through the urban matrix or use remnant vegetation patches without waterways. Consequently, populations persisting in wetland habitats now surrounded by urban infrastructure are essentially fragmented islands.

Within the northern cluster, Joondalup and Yanchep populations were less differentiated than were Joondalup and Herdsman. The Herdsman Lake population is closest to the city centre and has been within the urban footprint for the longest of all our study sites [70], suggesting that urban development has led to reduced gene flow between this population and the remaining northern populations, with resultant isolation and/or genetic drift. Herdsman Lake was naturally an ephemeral swamp, but since the 1850s, it has been subjected to stock grazing, market cropping, and attempted draining for land reclamation until it was finally dredged and

modified to be a compensation basin for urban drainage [71, 72]. As Perth's urban footprint has grown over the last two centuries, tiger snakes may have contracted from the surrounding inter-linking wetlands into the Herdsman Lake reserve. Yanchep and Joondalup wetlands are the northern and southern extremities of the Spearwood Dune System chain of lakes [73], and tiger snakes would have had historic population connectivity along this dune system. Urban development began around Joondalup Lake in the 1970s [70], and based on the current land-use (Fig 1) and our results, the Yanchep and Joondalup wetland populations may still be connected. However, continuing urban development around Joondalup Lake may result in this population developing similar genomic characteristics to the Herdsman Lake population in the near future.

As expected, Bibra and Kogolup lake populations were very closely related. These lakes are part of the Beeliar Regional Park, a chain of wetlands and woodlands currently managed as conservation land [74]. The Beeliar Regional Park is the only remaining connected wetland and woodland ecosystem in the Perth metropolitan area and should provide population connectivity for tiger snakes as long as it remains undeveloped. While there is likely to be some level of mortality due to the presence of arterial roads through the region, it appears there may still be some connectivity between these populations based on the close relationship shared by the Beeliar wetland populations and the Black Swan population.

Interestingly, the southern cluster exhibited a lack of spatial autocorrelation at 12 km, compared to 30 km in the northern cluster. While there appears to be a clear difference in the patterns between subregions, interpreting these differences is difficult without more detailed sampling at intervals of equal distance between populations. This is potentially very difficult to achieve as tiger snakes may not be present at many locations other than those already sampled in this study. The differences may also be partly the result of lower levels of genomic variation present in the northern cluster, with less variation leading to increased spatial autocorrelation of genotypes.

## Genomic diversity and health

As predicted, the northern cluster of tiger snakes had lower genomic diversity than the southern cluster, reflecting a similar pattern seen in quenda [35]. Yanchep is near the northern extent of the Western tiger snakes species range (Fig 1). As edge populations often harbour lower diversity than core populations [75], we suspect the relatively low diversity in the northern cluster is probably caused by the Swan/Canning river system isolating populations at the northern edge of the species range from gene flow and increasing genetic drift. Interestingly, genomic diversity was not lowest in the two sites with the highest genomic differentiation and geographic isolation, Herdsman Lake and Carnac Island, suggesting that isolation of ~90–150 years has not increased genetic drift in these populations.

Although none of the studied populations appeared to be strongly inbred, we found that $F_{IS}$ values closely reflected contemporary isolation of populations. For example, Carnac Island is insular and Herdsman Lake is isolated due to urbanisation and showed the highest signal of inbreeding, whereas the study sites with the most habitat connectivity (Yanchep, Kogolup and Black Swan) showed no signal of inbreeding. Inbreeding depression reduces individual fitness, survival and reproduction and can lead to rapid decline and extirpation of populations [17, 20, 76]. Despite small $F_{IS}$ values, population-level changes may not be seen for many generations in longer-lived vertebrates [18, 77]–such as tiger snakes, estimated at 10 years [78]. Thus, we expect to see inbreeding increase through time, especially in sites that become completely isolated from urbanisation.

Phenotypic signatures of inbreeding depression can be measured in wild populations using heterozygosity-fitness correlations. We found a strong correlation between inbreeding (*f*) and

heterozygosity, justifying our use of heterozygosity as a proxy for inbreeding. Our model found no effect of individual genomic heterozygosity on snake body condition, despite the broad pattern of the northern cluster sharing both lower heterozygosity and lower body condition (Fig 4). Body condition is a single measurement of fitness and low heterozygosity can translate to many other measures of fitness such as reduced body size in neonates [34], higher parasitism [79] and reduced survival probability [19]. Populations with low heterozygosity may be experiencing changes in other life history traits that could directly or indirectly affect fitness. Similar to Sovic et al. [52], our results show that body condition was strongly influenced by site. This suggests site-specific environmental stressors such as differences in prey availability [80], anthropogenic disturbance [81] or physiological changes from bioaccumulation of contaminants [32, 82, 83] are probably more important factors than heterozygosity for reducing body condition in tiger snakes from our study sites.

In addition, our use of genome-wide loci, possibly includes many loci in non-coding regions of the genome [84], which would possibly conceal the signal from SNPs under selection. The outlier test identified 131 loci that are potentially influencing fitness in different environments in the Perth region, and these candidate loci can be investigated in future analyses. However, as many traits that contribute to local adaptation are polygenic and may not exhibit high $F_{ST}$ values, we suggest that a genotype-environment association analysis would be a better method for investigating adaptive genomic architecture [85, 86]. Further investigation sampling tiger snake populations across their entire distribution–covering a range of native and urban wetland areas–would help identify SNPs that play a role in urban adaptation.

The above results demonstrate that contemporary genomic diversity in Western tiger snakes is more affected by population edge effects in conjunction with historical landscape isolation (Swan/Canning river playing the major role in population isolation and Yanchep at the northern edge experiencing lowered diversity) as opposed to fragmentation and isolation from urbanisation. The health of the northern SCP wetlands are continuously being threatened by anthropogenic water abstraction and climate change [87] in conjunction with ever-encroaching urbanisation [88]. Eventually, the larger urban wetlands (such as Herdsman Lake) will be the only islands of refuge for the northern cluster of tiger snakes. The northern SCP population already shows the lowest genomic diversity–hence adaptive potential–and poorest body condition, and thus is most likely at risk of future extirpation as urbanisation amplifies isolation as well as introducing novel environmental stressors.

## Effective population sizes

The largest estimates of $N_e$ came from the largest wetlands: Herdsman and Joondalup Lakes, despite these populations having relatively lower heterozygosity values and positive inbreeding coefficients. In contrast, the Black Swan population (the smallest lake) showed the lowest $N_e$ value despite high genomic diversity and no evidence of inbreeding. Rather than indicating isolation and genetic drift, our $N_e$ estimates appear to reflect the area and quality of available habitat at each locality. Similarly, Wood, Rose [65] found that despite high levels of isolation due to urbanisation, a population of the wetland snake *Thamnophis sirtalis tetrataenia* had the largest $N_e$ compared to other studied populations, suggesting that habitat restoration and enhancements may have facilitated high adult abundance at this locality. While Fraser, Ironside [89] suspect that large available habitat is responsible for maintaining large population sizes in deer populations isolated by urbanisation. Together, these results suggest that the quality and area of suitable habitat at our sampling sites is driving the effective population sizes of tiger snakes.

The $N_e$ estimate for Kogolup Lake population was infinite, and the lower parametric confidence interval was 3581 (Table 3). As our sample size for that population was $n = 10$, the infinite estimate is likely due to a small sample size; however, an infinite estimate can also mean there is no evidence for drift in that population, or the $N_e$ is actually large (>1000). Consequently, our dataset is unable to distinguish whether or not the population is 'very large'. The regional population structure, negative inbreeding coefficients, and current landscape connectivity between Black Swan, Bibra and Kogolup Lakes suggest that together these populations actually represents a broader Beeliar Regional Park meta-population, and we speculate that the infinite $N_e$ estimates could be a result of a large population and therefore not likely to suffer from genetic drift in the near future. Without increasing the sample size and recalculating the effective population size, we cannot confirm this population size. However, if the Kogolup Lake $N_e$ is actually large then the lower bound may provide useful information about plausible $N_e$ estimates [66, 90].

In conservation genetics, small population sizes limit adaptive potential [91]. Specifically, $N_e \geq 100$ is recommended to avoid inbreeding depression over the next five generations, while $N_e \geq 1000$ is recommended to maintain evolutionary potential; populations below this $N_e$ will suffer a reduced ability to evolve to cope with environmental change over time [63]. Four of our seven study populations are at an $N_e < 100$ (Table 3), and Joondalup and Herdsman lakes were not substantially higher than $N_e = 100$. Since most of our study populations are already showing signals of inbreeding, are completely isolated, or are in the process of becoming isolated due to urbanisation, ultimately all these populations are at risk of genetic degradation. If $N_e$ is strongly influenced by area and quality of available habitat, then habitat conservation, management and restoration may be the best method to buffer the further loss of genetic diversity in urban island tiger snake populations.

## On the origin of Carnac Island snakes

The Carnac Island tiger snakes showed a high level of genetic distinctiveness; this is not surprising given this population lives on an off-shore island. The Carnac Island population, unexpectedly also had the highest level of genomic diversity, and the highest frequency of private alleles (328 compared to 56–111 in mainland populations). For a population that was suspected to be introduced ~ 90 year ago, and shares ancestry with the geographically-closest populations of the southern cluster (Black Swan Lake), this is surprising. Considering the small size of Carnac Island (19 ha), we expected the tiger snake population to show low genomic diversity as island populations are renowned for having lower genetic diversity compared to adjacent mainland populations [92–94], even when the island introduction is less than 100 years [76]. A large founding population could have resulted in high heterozygosity [92, 95], however just ~ 40 adult snakes [36] were speculated to have been released on Carnac Island. Maintaining a large population size over time would also be necessary, as extended bottlenecks in the population size would have led to reductions in genetic diversity [95, 96]. It is possible that the founding population was sourced from many genetically diverse populations (e.g. including the east coast subspecies), if that was the case however, we would expect the Carnac Island snakes to separate from our sampled populations at lower $K$ values, and the geographically closest sampled populations to show little-to-no shared ancestry with Carnac Island in our admixture plots.

Surprisingly, we found the Carnac Island population had more than three to five times the private alleles compared to the mainland populations, much more than we would expect from *de novo* mutations over 90 years of isolation. We propose three hypotheses: (1) the snakes originated from other unsampled populations and the mutations are ancestral; (2) the mutation

rates have increased as a response to novel selection pressures [97], since the ecosystem on Carnac Island is very different to the habitat tiger snakes usually prefer on the mainland. This hypothesis could be supported by the phenotypic plasticity shown in the population [24], if epigenetically-driven plasticity has increased genome evolution [98]; or (3) the snakes are a naturally-occurring remnant population that is much older than 90 years, and the mutations are *de novo*. This hypothesis could be supported by Black Swan Lake's–the geographically-closest sampled population–shared ancestry, and tiger snakes naturally occurring on the nearby Garden Island [99], historically part of the land-bridge that connected these islands to the mainland roughly 6000 years ago [100].

## Conclusions

Urbanisation modifies ecosystems around the world, creating a range of stressors for wildlife living in cities. By investigating population genomic structure of species persisting in urban environments, we can gain useful information for conservation management of urban wildlife. Here, we genotyped urban and peri-urban populations of tiger snakes with the aim of understanding natural and anthropogenic influences on genomic diversity and population connectivity. We found that the major river system that runs through our urban study area has been a strong historical barrier to gene flow, resulting in the partial isolation of populations to the north of the river from the remainder of the species distribution. These northern populations also exhibited lower genomic diversity and lower body condition than southern populations, suggesting that they are most at risk of extirpation as urbanisation further encroaches upon their sensitive wetland habitats. As we expected, the populations most exposed to isolation– both geographic and urban–showed the strongest signal of inbreeding, although the maintenance of large effective population sizes appears to be driven primarily by the amount of available habitat. Together, these findings suggest that that increasing population connectivity and maximising the area of habitat in urban areas will help improve the adaptive capacity of urban wildlife. We also recommend further investigation into the genomic architecture of adaptation to urbanisation in this species, which will improve our understanding of the genetic and physiological pathways by which species adapt to urban environments.

## Supporting information

**S1 Fig. Hierarchical clustering dendrogram representing genetic distance relationships between *Notechis scutatus occidentalis* populations around Perth, Western Australia.** Calculated using 4688 single-nucleotide polymorphisms from across the genome. Note that these relationships do not necessarily reflect a true phylogeny.
(TIF)

**S2 Fig. Cross-entropy scree plot used to identify hierarchical population structuring in the genomic dataset for *Notechis scutatus occidentalis*.** Lower values of the cross-entropy criterion indicate a better fit to the data.
(TIF)

**S3 Fig. Admixture bar plot comparing population structure in *Notechis scutatus occidentalis* from Perth, Western Australia.** Each tick mark on the x-axis represents an individual snake, which are grouped by sampling locations. The dashed line represents the biogeographic barrier of the Swan/Canning Rivers separating the northern and southern sampling localities. The y-axis represents the fraction of individuals' genome that originates from a particular ancestral population, each of which has been given a unique colour.
(TIF)

**S4 Fig. Spatial autocorrelation of multilocus genotypes for** *Notechis scutatus occidentalis* **from Perth, Western Australia at five distance classes.** Cluster indicates the population used for analysis. Global is all mainland snakes, Northern and Southern are the populations either side of the Swan/Canning River system. The probability value at each distance class shows the proportion of permuted *r* values greater than the observed value in that distance class, based on 999 permutations of the SNP by sample matrix.
(TIF)

**S5 Fig. Hierarchical population genomic structure of** *Notechis scutatus occidentalis* **around Perth, Western Australia.** Unique colours in each panel represent an ancestral cluster. Black points indicate sampling sites. The dashed line represents the Swan/Canning River systems, while the dashed ring outlines Carnac and Garden Islands.
(TIF)

**S1 Table. Tukey HSD pairwise post-hoc test comparing body condition among six populations of** *Notechis scutatus occidentalis* **around Perth, Western Australia.**
(DOCX)

## Acknowledgments

We thank Jordan Vos, Kady Grosser, Serin Subaraj, Olimpia Cercora and Aleesha Turner for volunteering their time to help collect tissues for this project. We also pay respects to the traditional owners of the land, the Whadjuk-Noongar people, where this research was conducted.

## Author Contributions

**Conceptualization:** Damian C. Lettoof, Brenton von Takach.

**Data curation:** Damian C. Lettoof, Vicki A. Thomson, Jari Cornelis, Fabien Aubret, Brenton von Takach.

**Formal analysis:** Damian C. Lettoof, Brenton von Takach.

**Funding acquisition:** Damian C. Lettoof, Vicki A. Thomson.

**Investigation:** Damian C. Lettoof, Brenton von Takach.

**Methodology:** Damian C. Lettoof, Brenton von Takach.

**Project administration:** Damian C. Lettoof.

**Resources:** Damian C. Lettoof.

**Supervision:** Philip W. Bateman, Fabien Aubret, Marthe M. Gagnon, Brenton von Takach.

**Visualization:** Damian C. Lettoof, Brenton von Takach.

**Writing – original draft:** Damian C. Lettoof, Brenton von Takach.

**Writing – review & editing:** Damian C. Lettoof, Vicki A. Thomson, Jari Cornelis, Philip W. Bateman, Fabien Aubret, Marthe M. Gagnon, Brenton von Takach.

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
