## [Decision Letter · Decision Letter 0]

10 Sep 2021

PONE-D-21-24399Top predator snake shows genomic signatures of natural and anthropogenic barriers to gene flowPLOS ONE

Dear Dr. Lettoof,

Thank you for submitting your manuscript to PLOS ONE. After careful consideration, we feel that it has merit but does not fully meet PLOS ONE’s publication criteria as it currently stands. Therefore, we invite you to submit a revised version of the manuscript that addresses the points raised during the review process.

We look forward to receiving your revised manuscript.

Kind regards,

Randeep Singh

Academic Editor

PLOS ONE

Journal Requirements:

2. In your Methods section, please provide additional location information about your sampling sites, including geographic coordinates for the data set if available.

4. We note that Figure 1 in your submission contain map images which may be copyrighted. All PLOS content is published under the Creative Commons Attribution License (CC BY 4.0), which means that the manuscript, images, and Supporting Information files will be freely available online, and any third party is permitted to access, download, copy, distribute, and use these materials in any way, even commercially, with proper attribution. For these reasons, we cannot publish previously copyrighted maps or satellite images created using proprietary data, such as Google software (Google Maps, Street View, and Earth). For more information, see our copyright guidelines: http://journals.plos.org/plosone/s/licenses-and-copyright.

Reviewers' comments:

Reviewer's Responses to Questions

**Comments to the Author**

1. Is the manuscript technically sound, and do the data support the conclusions?

Reviewer #1: Yes

Reviewer #2: Yes

2. Has the statistical analysis been performed appropriately and rigorously? 

Reviewer #1: Yes

Reviewer #2: Yes

3. Have the authors made all data underlying the findings in their manuscript fully available?

Reviewer #1: Yes

Reviewer #2: Yes

4. Is the manuscript presented in an intelligible fashion and written in standard English?

Reviewer #1: Yes

Reviewer #2: Yes

5. Review Comments to the Author

Reviewer #1: 1. ORIGINAL SUBMISSION

1.1. Recommendation Accepted with minor revision

2. COMMENTS TO AUTHOR

Authors: Damian C. Lettoof, Vicki A. Thomson, Jari Cornelis, Philip W. Bateman, Fabien Aubret, Marthe M. Gagnon, Brenton von Takach

Title: Top predator snake shows genomic signatures of natural and anthropogenic barriers to gene flow

2.1. Overview and general recommendation:

This study describes the effects of urbanization on tiger snakes using population genomics around the Perth region in Australia to access and inform conservation strategies. For this study, the authors used RAD-associated SNPs to evaluate the genetic structure among populations of Notechis scutatus occidentalis of different localities along with its distribution. This article is well-written, with a few issues related to interpretation that need to be considered. Therefore, I accepted this manuscript for publication, but with minor revision. Please, find below my comments and consideration of this manuscript.

I think that the results about the genetic distance among populations are over-interpreted. Although Carnac has the greatest genetic distance among other populations (0.40-0.29), the genetic distance between Carnac and Black Swan/Kogolup is moderate (0.29). Also, it is moderate when the genetic distance is compared among distant populations, for example, Yanchep and Black Swan (0.26). So, I conclude that the genetic distance among populations is not high but moderate to low, and they are not entirely isolated.

About the investigation of the cross-entropy scree plot produced using the tess3r package, in Figure 3, I do not see the plot of K=4 in the figure. I just saw the plot of K=2 to K=3. Where is this plot (K=4)? In Figure S3, I just saw the plot of K=5 to K=7. Also, in Figure 3 and Figure S3, I suggest that the south and north lakes should be indicated in these figures because it is tough to understand what the authors present in results and what they have shown in these figures.

Also, following the cross-validation interpretation, to choose the best K, in Figure S2, the author said in Materials and Methods: “We considered ancestry coefficients for K values of 2, 3 and 4 as these values showed the largest improvements in cross-entropy, with K = 4 showing a substantial change in slope, and plotted these for each individual”.

In the manual of tess3r they mentioned: “The best choice for the K value is when the cross-validation curve exhibits a plateau or starts increasing… Be cautious about over-interpreting the value of K and the folkore around the choice of this value. Population structure is often hierarchical, and the estimation of K strongly depends on sampling and genotyping efforts. The number of genetic groups detected by ancestry estimation programs does not necessarily correspond to the number of biologically meaningful populations in the sample (Francois and Durand 2010). If I am correct, following the Figure S2 the best K in your analysis is K=4 to K=6, so why you choose K=2 to K=4?

Also, the author mentioned, “At K=3, Herdsman Lake is recovered as a distinct cluster, potentially due to isolation from a sea of urban infrastructure”. I do not see this in figure 3, I see a strong differentiation, but I see that there is gene flow with other populations. I suggest that the authors use additional analysis to see gene flow, as Ima2 and FastSimCoal. Also, the author can use Estimated Effective Migration Surface analysis (EEMS); in this analysis, you can see the landscape and the genetic structure among populations. Also, you can evaluate some geographic barriers that can explain the population structure.

Reviewer #2: A comprehensive study that makes important theoretical contributions as well as using its data to guide management action for conservation.

I have only minor comments to make.

Line 2: "Top predator" probably not justifiable, nor necessary in title, or elsewhere in the MS. No evidence that tiger snakes are top or apex predators is presented in the MS or cited references. Seems to just be words added to colour the MS.

Line 19: Abstract. Discovery of the much greater than predicted heterozygosity on Carnac was not an aim of the study but becomes a significant finding. It should be mentioned in the Abstract.

Line 30: why the quotes around ‘fitness’?

Line 95: Study Sites and Sample Collection. In this section could the authors comment on the unevenness of the sampling at the different sites? Low samples at some sites could reflect low population numbers, but other factors such as snake detectability, habitat complexity or access could also influence sampling. This unevenness, leading to potential population underestimates, could then be part of the potential set of explanations for differences between sites.

This topic could be part of the Discussion around population size estimates in lines 429-450

Line 202: again, why the quotes?

Line 383: isolated due to urbanisation

Line 390: and reduced survival probability

Line 425-26: incomplete sentence: While Fraser, Ironside….habitat.

Line 439: typo: Kogolup

Lines 466-67: Awkward sentence structure – needs rewrite.

6. PLOS authors have the option to publish the peer review history of their article (what does this mean?). If published, this will include your full peer review and any attached files.

Reviewer #1: No

Reviewer #2: No

---

## [Author Response · Author response to Decision Letter 0]

22 Sep 2021

Reviewer 1

I think that the results about the genetic distance among populations are over-interpreted. Although Carnac has the greatest genetic distance among other populations (0.40-0.29), the genetic distance between Carnac and Black Swan/Kogolup is moderate (0.29). Also, it is moderate when the genetic distance is compared among distant populations, for example, Yanchep and Black Swan (0.26). So, I conclude that the genetic distance among populations is not high but moderate to low, and they are not entirely isolated.

These values are not genetic distance, they are genetic differentiation (based on G″ST, an FST analogue). As the number of migrants strongly influences the level of differentiation, we can conclude that the population with the highest level of differentiation from other populations is likely to be the most isolated. We suggest that a differentiation value of 0.4 indicates quite strong isolation from other populations. However, we agree that we can express the results in a more communicable fashion, and have now revised the relevant paragraph in the Results section to state:

“Carnac Island was the most differentiated from other populations (pairwise G″ST = 0.29–0.40). The most geographically distant pair of sites, Yanchep and Black Swan, exhibited a moderate level of differentiation (G″ST = 0.26).”

About the investigation of the cross-entropy scree plot produced using the tess3r package, in Figure 3, I do not see the plot of K=4 in the figure. I just saw the plot of K=2 to K=3. Where is this plot (K=4)? In Figure S3, I just saw the plot of K=5 to K=7. Also, in Figure 3 and Figure S3, I suggest that the south and north lakes should be indicated in these figures because it is tough to understand what the authors present in results and what they have shown in these figures.

We accidently submitted the old figure with incorrect labelling. Fig 3 shows K2, 3 & 4. We have attached the correct version, and added additional labels clarifying the populations north/south of the rivers. We also state in the figure caption that the dashed line is the divide between north/south populations. 

Also, following the cross-validation interpretation, to choose the best K, in Figure S2, the author said in Materials and Methods: “We considered ancestry coefficients for K values of 2, 3 and 4 as these values showed the largest improvements in cross-entropy, with K = 4 showing a substantial change in slope, and plotted these for each individual”.

In the manual of tess3r they mentioned: “The best choice for the K value is when the cross-validation curve exhibits a plateau or starts increasing… Be cautious about over-interpreting the value of K and the folkore around the choice of this value. Population structure is often hierarchical, and the estimation of K strongly depends on sampling and genotyping efforts. The number of genetic groups detected by ancestry estimation programs does not necessarily correspond to the number of biologically meaningful populations in the sample (Francois and Durand 2010). If I am correct, following the Figure S2 the best K in your analysis is K=4 to K=6, so why you choose K=2 to K=4?

With large genomic datasets utilising thousands of SNPs, the K value with the lowest cross-entropy is typically the same value as the overall number of sampling localities, as we have the power to identify even fine-scale differentiation among geographically close sites. This is not very informative because it doesn’t group populations into ancestral clusters or allow for inference of meaningful population structure across the landscape. A large drop in cross-entropy implies a big improvement in the explanation of genetic structure and so is useful for inference about higher levels of the hierarchy. We have now reworded the relevant paragraph of the Materials and Methods section to improve our explanation of the method, which now reads:

“The ‘tess3’ function incorporates the latitudes and longitudes of each sampled individual to account for the influence of isolation-by-distance on ancestry coefficients, with large drops or plateaus in the scree plot identifying useful values of K for inference of population genetic structure (Fig S2). We considered K values of 2, 3 and 4 most useful for describing the high-level genetic structure across the region, as these values showed the largest reductions in cross-entropy, with a plateau starting to form at K > 4. For finer-scale investigation of population structuring we also plotted ancestry coefficients of K = 5, 6 and 7 (Fig S3).”

Also, the author mentioned, “At K=3, Herdsman Lake is recovered as a distinct cluster, potentially due to isolation from a sea of urban infrastructure”. I do not see this in figure 3, I see a strong differentiation, but I see that there is gene flow with other populations. I suggest that the authors use additional analysis to see gene flow, as Ima2 and FastSimCoal. Also, the author can use Estimated Effective Migration Surface analysis (EEMS); in this analysis, you can see the landscape and the genetic structure among populations. Also, you can evaluate some geographic barriers that can explain the population structure.

Our apologies for inserting an old figure. The reviewer was thus interpreting these findings on K=2, not K=3. The new figure should improve understanding in this regard. To address the issue of visualising the pattern of population structure across the landscape, we have now included maps of the ancestry coefficients output from tess3r, for K values of 2 to 6, in the Supplementary Material. We believe this adequately allows readers to visually identify barriers to gene flow.

Reviewer 2

Line 2: "Top predator" probably not justifiable, nor necessary in title, or elsewhere in the MS. No evidence that tiger snakes are top or apex predators is presented in the MS or cited references. Seems to just be words added to colour the MS.

Although there is no formal study quantifying the trophic tier for tiger snakes in Australia, they are well recognised by herpetolologists - including very experienced authors of this MS – as not, or rarely, suffering predation once they attain adult size. This is supported by the lack of adult predation records in the literature. As an alternative, we propose changing the phrase in the title to “Bioindicator snake” – as there are currently six publications justifying the use of tiger snakes as bioindicators (see Lettoof, D. C. et al.).

Line 19: Abstract. Discovery of the much greater than predicted heterozygosity on Carnac was not an aim of the study but becomes a significant finding. It should be mentioned in the Abstract.

Our abstract is currently at the word limit and we believe it adequately summarises the key findings. Although the Carnac discovery was interesting, it was not a primary focus of the MS and we cannot describe the findings – with appropriate context – in an already filled abstract.

Line 30: why the quotes around ‘fitness’?

We have removed these quotes.

Line 95: Study Sites and Sample Collection. In this section could the authors comment on the unevenness of the sampling at the different sites? Low samples at some sites could reflect low population numbers, but other factors such as snake detectability, habitat complexity or access could also influence sampling. This unevenness, leading to potential population underestimates, could then be part of the potential set of explanations for differences between sites.

This topic could be part of the Discussion around population size estimates in lines 429-450

Kogolup, Black Swan and Carnac Island were opportunistically sampled to add sites to this study, and received less survey effort than Herdsman, Joondalup, Bibra and Yanchep. We are confident that our detection ability in catchers was the same at every site, but the lower sample size was related to survey effort. We have added a sentence in the method to clarify “Kogolup Lake, Black Swan Lake and Carnac Island were surveyed less than the other sites (a few days compared with several weeks), which resulted in lower sample sizes for these sites.”

Line 202: again, why the quotes?

We have removed these quotes.

Line 383: isolated due to urbanisation

We have amended this sentence.

Line 390: and reduced survival probability

We have added ‘reduced’.

Line 425-26: incomplete sentence: While Fraser, Ironside….habitat.

We have restructured the sentence to “While Fraser, Ironside (88) suspect that large available habitat is responsible for maintaining large population sizes in deer populations isolated by urbanisation.”

Line 439: typo: Kogolup

Amended.

Lines 466-67: Awkward sentence structure – needs rewrite.

We have rewritten to “It is possible that the founding population was sourced from many genetically diverse populations (e.g. including the east coast subspecies), if that was the case however, we would expect the Carnac Island snakes to separate from our sampled populations at lower K values, and the geographically closest sampled populations to show little-to-no shared ancestry with Carnac Island in our admixture plots.”

---

## [Editor Report · Decision Letter 1]

13 Oct 2021

Bioindicator snake shows genomic signatures of natural and anthropogenic barriers to gene flow

PONE-D-21-24399R1

Dear Damian ,,

In the manuscript authors have fairly done the good work and have scientific merit. My suggestions is to accept the manuscript after incorporating the suggestions.

Kind regards,

Randeep Singh

Academic Editor

PLOS ONE

Additional Editor Comments (optional):

In the manuscript authors have fairly done the good work and have merit. My suggestions is to accept the manuscript after incorporating the suggestions.
---

## [Editor Report · Acceptance letter]

21 Oct 2021

PONE-D-21-24399R1 

Bioindicator snake shows genomic signatures of natural and anthropogenic barriers to gene flow 

Dear Dr. Lettoof:

I'm pleased to inform you that your manuscript has been deemed suitable for publication in PLOS ONE. Congratulations! Your manuscript is now with our production department. 

Kind regards, 

on behalf of

Dr. Randeep Singh 

Academic Editor

PLOS ONE